# Ion Signaling in Cell Motility and Development in *Dictyostelium discoideum*

**DOI:** 10.3390/biom14070830

**Published:** 2024-07-10

**Authors:** Yusuke V. Morimoto

**Affiliations:** 1Faculty of Computer Science and Systems Engineering, Kyushu Institute of Technology, 680-4 Kawazu, Iizuka 820-8502, Fukuoka, Japan; yvm001@phys.kyutech.ac.jp; 2Japan Science and Technology Agency, PRESTO, 4-1-8 Honcho, Kawaguchi 332-0012, Saitama, Japan

**Keywords:** signal transduction, ion signaling, calcium signaling, intracellular pH, *Dictyostelium discoideum*, cell motility, cell differentiation

## Abstract

Cell-to-cell communication is fundamental to the organization and functionality of multicellular organisms. Intercellular signals orchestrate a variety of cellular responses, including gene expression and protein function changes, and contribute to the integrated functions of individual tissues. *Dictyostelium discoideum* is a model organism for cell-to-cell interactions mediated by chemical signals and multicellular formation mechanisms. Upon starvation, *D. discoideum* cells exhibit coordinated cell aggregation via cyclic adenosine 3′,5′-monophosphate (cAMP) gradients and chemotaxis, which facilitates the unicellular-to-multicellular transition. During this process, the calcium signaling synchronizes with the cAMP signaling. The resulting multicellular body exhibits organized collective migration and ultimately forms a fruiting body. Various signaling molecules, such as ion signals, regulate the spatiotemporal differentiation patterns within multicellular bodies. Understanding cell-to-cell and ion signaling in *Dictyostelium* provides insight into general multicellular formation and differentiation processes. Exploring cell-to-cell and ion signaling enhances our understanding of the fundamental biological processes related to cell communication, coordination, and differentiation, with wide-ranging implications for developmental biology, evolutionary biology, biomedical research, and synthetic biology. In this review, I discuss the role of ion signaling in cell motility and development in *D. discoideum*.

## 1. Introduction

Cell-to-cell communication is essential in multicellular organisms, facilitating organization and allowing them to function as a single multicellular system [1,2]. Cell signals are received by the plasma membrane receptors of other cells, which are then transduced into intracellular signals, resulting in cellular responses such as changes in gene expression and protein function. Signal propagation within multicellular systems facilitates the expression of integrated functions in individual tissues. Signal-regulated, collective cell migrations have a wide range of functions and contribute to several processes such as morphogenesis, wound healing, and cancer invasion [3]. Signaling molecules include proteins, such as G proteins; peptides; and low-molecular-weight chemicals, such as nucleotides or even ions [4,5,6]. Advantageously, ion-mediated signals, such as action potentials in neurons, can be transmitted rapidly and over long distances [7,8]. Furthermore, the cell membrane potential comprises the total ion concentration gradient inside and outside the membrane, as expressed by the Nernst equation. The ion concentration gradient is established by selective ion transporters, including ion pumps, and ion channels facilitate the flow of specific ions down that gradient [9]. Accordingly, changes in the concentration of various ions act as signals. For instance, calcium ions act as second messengers and are essential signals for a wide range of biological functions, including muscle contraction, exocytosis, neurotransmission, gene expression, fertilization, and cell growth [10,11,12]. Unlike biosynthesized signals, ions are absorbed from the environment, making them versatile and essential signaling factors for organisms; these characteristics are widely conserved from bacteria to mammalian cells [13,14,15,16,17,18].

The cellular slime mold, *Dictyostelium discoideum*, is a model organism for signal transduction because of its widely conserved chemical signal-mediated cell-to-cell interactions and multicellular formation mechanisms [19,20,21]. *D. discoideum* cells normally grow and multiply as unicellular organisms, but upon starvation, they aggregate and undergo a transition from a unicellular to multicellular organism (Figure 1) [19,20]. In such cases, approximately 100,000 cells aggregate to form a single multicellular body. This coordinated cell migration is mediated by the self-formation of cyclic adenosine 3′,5′-monophosphate (cAMP) gradients and chemotaxis to extracellular cAMP [22]. When cells sense extracellular cAMP, adenylate cyclase is activated via G proteins and cAMP is synthesized intracellularly [23,24]. The synthesized cAMP is then secreted extracellularly, and neighboring cells respond in a similar manner, resulting in an intercellular cAMP signal relay. cAMP relays propagate as a traveling wave and achieve collective migration by chemotaxis to the aggregation center [19]. At this time, the calcium signaling occurs synchronously with the cAMP signaling [25]. The formed multicellular body, also called a slug, moves in one direction, entirely facilitated by organized collective migration. Differentiation-inducing factor 1 (DIF-1) and bis-(3′-5′)-cyclic dimeric guanosine monophosphate (cyclic-di-GMP) contribute to the spatiotemporal differentiation pattern regulation within the multicellular body as inducers of differentiation toward the stalk cells [26,27,28]. Eventually, a fruiting body consisting of a stalk and spores is constructed [21]. Additionally, cAMP contributes to the differentiation of both the stalk and the spores [21]. Upon reaching a suitable environment, the spores germinate and return to the unicellular amoeboid state. In this life cycle, the cells are in the haploid phase, which facilitates genetic manipulation (such as gene disruption) and phenotypic changes. *D. discoideum* has long been studied due to its simple life cycles, which include multicellularity, easy culturing, and genetic manipulation. Furthermore, it shares highly conserved proteins and signaling pathways with higher eukaryotes [29]. This review provides an overview of ion signaling during cell-to-cell signaling and the differentiation processes in multicellular formation in *Dictyostelium* cells.

## 2. Ion Signals in *Dictyostelium*

### 2.1. Calcium Signals

Along with the cAMP signaling pathway, which is well studied in *D. discoideum*, the Ca^2+^ signaling pathway is a widely conserved signaling system in biology. Ca^2+^ functions as a second messenger despite being a simple metal ion [5,12]. The steady-state cytosolic Ca^2+^ concentration is kept low (in the nM range), but upon stimulation, the cytosolic Ca^2+^ concentration increases rapidly to several hundred nM, leading to various cellular responses such as gene expression and cell differentiation [10,30]. The cytosolic Ca^2+^ concentration increases are facilitated by two main pathways: extracellular influx and intracellular Ca^2+^ store release. The endoplasmic reticulum (ER) is largely responsible for the intracellular Ca^2+^ storage. The intra- and extracellular Ca^2+^ pathways also play a role in signaling in *D. discoideum* cells [25,31,32,33,34,35,36,37].

#### 2.1.1. Ca^2+^ in Chemotaxis and Cell Motility

In the cAMP signaling relay of *D. discoideum* cells, the cytosolic Ca^2+^ concentration transiently increases in response to cAMP stimulation [38,39,40]. Therefore, during the aggregation phase, periodic oscillations in the cytosolic Ca^2+^ concentration, similar to the cAMP relay, are observed [25,41,42]. The transient increase in the cytosolic Ca^2+^ concentration is observed at different developmental stages as a response to the cAMP receptors, cAR1, cAR2, and cAR3, and potentially via G protein-independent pathways [43,44]. Otherwise, Ca^2+^ elevation occurs in response to folate [34], DIF1 and DIF2 [45], cyclic-di-GMP [27], L-glutamate, gamma-aminobutyric acid (GABA) [46], and the polyketide, 4-methyl-5-pentylbenzene-1,3-diol (MPBD) [47]. Transient increases in cytosolic Ca^2+^ concentrations also occur in response to ATP and ADP via the polycystin-2 homolog, TrpP (also known as PKD2) [45]. ATP- and ADP-mediated calcium signaling have also been implicated in P2X receptors localized in the contractile vacuole required for osmoregulation [48,49,50,51]. Transient increases in Ca^2+^ levels occur in response to different stimuli. However, since the receptors that receive these signals have different specificities, the cells can distinguish between the stimuli by utilizing distinct downstream signaling pathways and by varying the timing, duration, and location of the Ca^2+^ increase. Although Ca^2+^ signals are not essential for chemotaxis [52], disruptions in the cytosolic Ca^2+^ fluctuations can interfere with cAMP signal oscillations, suggesting a regulatory role of Ca^2+^ oscillation signals in modulating periodic signals [53,54,55,56,57]. Additionally, increases in the cytosolic Ca^2+^ concentration regulate cellular migration via myosin II heavy chain phosphorylation [58]. Furthermore, Ca^2+^ is implicated in cGMP signaling [59,60,61,62,63,64], suggesting it plays a role in controlling cell motility. Ca^2+^ elevation induced by cAMP stimulation exhibits photosensitivity, as exposure to 405 nm light significantly inhibits the increase in Ca^2+^ levels [65]. The dynamics of cytosolic Ca^2+^ oscillations and detectable Ca^2+^ elevations are primarily mediated by IplA, which is the only *Dictyostelium* ortholog of the mammalian IP3 receptor, a ligand-gated Ca^2+^ channel that releases Ca^2+^ from ER stores [55,57,66,67]. A contribution from the plasma membrane Ca^2+^ ATPase [68] has also been suggested. Acidic vesicles, including contractile vacuoles, contribute to Ca^2+^ signaling [69], with PAT1 in contractile vacuoles contributing to Ca^2+^ regulation [70]. Furthermore, the influx of Ca^2+^ from the extracellular environment is essential for galvanotaxis [71]. In strains lacking RegA, a cAMP phosphodiesterase, the basal concentration of intracellular Ca^2+^ increases, indicating the involvement of cAMP in regulating cytosolic Ca^2+^ levels [72]. Extracellular Ca^2+^ plays a role in the formation of cell polarity, with an optimal extracellular concentration of 10 mM [73]. Calcium chemotaxis in response to gradients of extracellular Ca^2+^ concentration is also observed, with contributions from the IP3 receptor homolog, IplA, and myosin heavy chain kinase [57,74].

#### 2.1.2. Ca^2+^ Signaling during Development

Intra- and extracellular Ca^2+^ concentrations influence *D. discoideum* cell development [75,76,77,78,79,80,81,82,83]. Ca^2+^ signaling pathways responsive to external stimuli undergo changes during development [25,84]. In the multicellular slug stage, transient increases in the cytosolic Ca^2+^ concentration in response to cAMP stimulation or mechanical stimulation are more pronounced in anterior prestalk cells than in posterior prespore cells [25,39,85], suggesting differences in Ca^2+^ storage capacity [86]. Calcium also affects slug migration and transiently increases the speed of slug migration after Ca^2+^ firing in response to mechanical stimuli [25]. Extracellular Ca^2+^ chelation disrupts slug phototaxis and thermotaxis [87]. DIF-1 stimulation induces an increase in cytosolic Ca^2+^ concentrations in prestalk cells [45,88], which in turn induces *ecmB* expression specifically in prestalk cells [77,89,90,91]. Conversely, Ca^2+^ is also required for prespore cell differentiation [78], with the calcium-dependent phosphatase calcineurin contributing to prestalk and prespore cell differentiation [92]. Calcium-binding proteins with distinct expression patterns in prestalk and prespore cells have been identified [93]. During sporulation, the cytosolic Ca^2+^ concentration decreases, but it increases during germination, contributing substantially to spore development [94], which is possibly related to spore-specific actin rod regulation [95]. Additionally, actin dynamics are regulated by annexin VII, which acts as a voltage-dependent Ca^2+^ channel, contributing to intracellular Ca^2+^ homeostasis during differentiation [96,97]. The adhesive factor, DdCAD-1, acts as an extracellular Ca^2+^-dependent adhesion molecule, contributing to multicellular morphogenesis [98,99,100].

#### 2.1.3. Ca^2+^ Signaling in Mechanosensation 

Ca^2+^ signaling plays a notable role in mechanosensation mechanisms across a wide range of organisms, from humans and plants to bacteria [101,102,103,104,105]. Mechanosensitive calcium responses in *D. discoideum* cells differ mechanistically from responses to chemical stimuli [106]. In *D. discoideum* cells, shear flow-induced mechanosensitive mechanotransduction involves the TrpP channel, which is a homolog of the Trp (transient receptor potential) channel family [45,107]. This channel family plays a role in receiving temperature, chemical, and mechanical stimuli, and is widely conserved in vertebrates [102]. Additionally, Ca^2+^ firing induced by a mechanical response occurs when the cell-attached substrate is pulled in *D. discoideum* cells [108]. In the unicellular stage, mechanosensitive responses induced by stimuli, such as agar covering, primarily involve an extracellular Ca^2+^ influx via Piezo homologs, promoting bleb motility rather than pseudopod motility [37]. Piezo channels are conserved in humans and play a role in the mechanical stimulus responses in different organs [101,103]. In contrast, in multicellular slugs, the contribution of Piezo homologs to the Ca^2+^ signal that follows mechanical stimuli is small, and the combination of *iplA* deficiency and chelator EGTA treatment, which inhibits the extracellular Ca^2+^ pathway, finally abolishes the Ca^2+^ response, indicating that not only extracellular Ca^2+^ but also Ca^2+^ flux from the ER is at work [25]. EGTA treatment results in a slower peak, suggesting that the mechanical response in multicellular bodies functions as a combination of a fast Ca^2+^ response from the extracellular vesicles and a slow Ca^2+^ response from the intracellular vesicles [25] (Figure 2). Despite the differences in signal pathways responding to mechanical stimuli between the unicellular and multicellular stages in the same strain of *D. discoideum*, the involvement of the same Ca^2+^ signals persists [25], providing intriguing insights into the evolution of mechanosensing mechanisms containing widely conserved Ca^2+^ signals.

#### 2.1.4. Determining Ca^2+^ Signals 

Various methods have been used to measure Ca^2+^ signals in *D. discoideum* cells [109]. Ca^2+^ influx and efflux were measured using radioisotopes in the past [38,43,84], but fluorescence-based determinations using probes, such as Fura-2, have recently provided significant spatiotemporal dynamic insights [40,72,110]. Measurements using the protein Ca^2+^ probe, aequorin, eliminated the need for time-consuming and invasive dye introduction, but the sensitivity of aequorin was not ideal for *D. discoideum* cells [85,111]. These cells fluctuate in Ca^2+^ levels in a lower concentration range than mammalian cells. In contrast, fluorescent protein-based Ca^2+^ probes have improved the sensitivity and temporal resolution of measurements. For instance, the fluorescence resonance energy transfer (FRET) sensor, Yellow Cameleon-Nano (YC-Nano) 15 [42], and GCaMP6s [25] enable time-lapse measurements of the calcium signal with high sensitivity. FRET probes are highly quantitative because of their ratio measurement capability, while the GCaMP series, for example, which can measure at a single wavelength, can reduce phototoxicity in time-lapse measurements because of the amount of excitation light irradiation. YC-Nano15, with a low K_d_ (K_d_ = 15 nM), is suitable for capturing Ca^2+^ fluctuations at low concentrations in the early stages of cell aggregation. On the other hand, GCaMP6s (K_d_ = 144 nM) is suitable for capturing a broad range of Ca^2+^ concentrations after the stream stage, when cell development has progressed and the Ca^2+^ concentration fluctuations have increased [25]. Thus, understanding the characteristics of each probe before its use is essential. To further elucidate the molecular dynamics of Ca^2+^ signaling, it is necessary to improve the spatial resolution and measure the fluctuations at the cellular and organelle levels.

### 2.2. pH Signals

In living cells, the selective movement of H^+^ and H_3_O^+^ leads to fluctuations in intracellular pH. From bacteria to human cells, the cytosolic pH is maintained at weak alkaline levels of approximately 7.1–7.5 [112,113]. The cytosolic pH of *D. discoideum* cells is approximately 7.2–7.5 [114,115,116,117,118], and its maintenance heavily relies on proton ATPases, which function as proton pumps by utilizing the energy from ATP hydrolysis [118,119]. *D. discoideum* cells possess several proton ATPases, which also contribute to plasma membrane potential maintenance [120]. In *D. discoideum* cells, proton ATPases are abundant in acidic vesicles [121,122,123], as are Ca^2+^/H^+^ ATPases [124,125].

#### 2.2.1. pH in Cell Motility 

In chemotaxis and cell migration, cellular motility at the single-cell level in *D. discoideum* is dependent on the intracellular pH [117]. A decrease in the intracellular pH of approximately 0.2 units has a minor impact on random migration but notably decreases the migration speed of chemotaxis motility. Further decreasing the intracellular pH also reduces the migration speed of random migration. On the other hand, increasing the intracellular pH through the addition of methylamine increases the migration speed of random migration but has little effect on chemotaxis motility [117]. pH influences motility by regulating pH-dependent actin-binding protein functions, thereby modulating actin function [126,127,128,129]. The Na^+^/H^+^ exchanger Nhe1, which controls the intracellular pH, localizes to the leading edge of chemotaxing cells. In *nhe1*-deficient strains, the localization of F-actin at the leading edge of the cell is markedly reduced, indicating that pH is elevated at the polarized leading edge of migrating cells to facilitate efficient chemotaxis [130]. Additionally, cAMP stimulation transiently elevates the cytosolic pH [131,132], resulting in pH oscillations in dense cell suspensions [133,134]. These oscillations are likely caused by the release of H^+^ when Ca^2+^ is taken up in response to chemical stimuli [131,135].

#### 2.2.2. pH Signaling during Development

During developmental processes, researchers have measured fluctuations in the intracellular pH during the cell cycle and differentiation processes of *D. discoideum* [114,132,136,137]. These measurements suggest a correlation between pH and cell differentiation [77,138,139,140]. Changes in the intra- and extracellular pH also impact gene expression [140,141]. In the multicellular stage, raising the intracellular pH using ionophores leads to the formation of elongated slugs [142]. The formation of fruiting bodies begins with the differentiation of prestalk cells into stalk cells. Prestalk cells are prone to pH reduction, and the presence of weak acids promotes their differentiation into stalk cells [143]. As a result, weak acids promote fruiting body formation, while weak bases, such as ammonia, which is produced during development, inhibit it [144,145]. The weak base ammonia inhibits developmental processes by raising the pH of acidic vesicles instead of the cytosol [146]. Differentiation-mediated self-organization patterns observed in two-dimensional (2D) cell cultures are regulated by pH and ammonia [147,148,149]. When prestalk cells are stained with neutral red, it suggests the development of acidic vesicles in these cells. The pH of the vacuoles in prestalk cells is notably lower than that of prespore cells [150,151], possibly indicating their involvement in the switch to stalk cell differentiation. When an extracellular pH gradient is created, slugs and fruiting bodies tend to orient toward the acidic side, which correlates with the orientation of prestalk cells in slugs [152,153].

#### 2.2.3. Measurement Techniques for Intracellular pH

The intracellular pH of *D. discoideum* cells has been measured using multiple techniques, including the use of the radioactive isotope tritium [151,154,155,156] and fluorescent dyes [116,136,137,157,158]. Furthermore, high spatiotemporal resolution measurements of the intracellular pH have been performed by measuring and visualizing it using pH-sensitive fluorescent proteins [118,130]. During the cell aggregation phase, a localized increase in pH at the leading edge of cells has been suggested, supported by the localization of NHE [130]. The relationship between NHE and intracellular pH is particularly important in cancer cell research because cancer cells have a higher intracellular pH than normal cells due to NHE activation. This suggests that intracellular pH elevation enhances disease symptoms in cancer cells [159,160]. Thus, measuring the behavior of intracellular pH at a high spatiotemporal resolution can provide insights into the molecular mechanisms of cellular function [113,161].

### 2.3. K^+^, Na^+^, and Fe^2+^ Signals and Membrane Potential

In the regulation of neuronal membrane potential signals, the selective movement of K^+^ and Na^+^ ions plays a significant role [7,8,9]. The involvement of Nhe1 in the polarity formation of aggregating *D. discoideum* cells suggests that extracellular K^+^ and Na^+^ concentrations also influence this process [73,162]. Furthermore, stimulation of cAMP triggers the efflux of K^+^ ions, which is dependent on Ca^2+^ [163]. It has been proposed that oscillations in K^+^ concentrations are associated with cAMP signaling [164]. The cAMP-induced K^+^ release is hindered by potassium channel blockers, indicating that certain potassium channels are activated in a cAMP relay-dependent manner [164]. When *D. discoideum* cells were developed on agar supplemented with potassium channel blockers, the stalks were more than twice as long as those without the blockers, suggesting that potassium channels contribute to development [165]. The intracellular Na^+^ concentration in *D. discoideum* cells has been measured using nuclear magnetic resonance (NMR), revealing a range of 0.6–3 mM despite extracellular levels ranging between 20 and 70 mM, indicating the maintenance of concentration gradients across the cell membrane [115]. K^+^ and Mg^2+^ also influence myosin function, contributing to cellular motility [166]. While the cortical localization of myosin II is crucial for cell polarity during cell migration, it is regulated by extracellular Ca^2^+. However, at an external concentration of 40 mM, K^+^ can partially substitute for Ca^2+^ [73]. Under these circumstances, the presence of Nhe1 is necessary for polarity formation in the presence of K^+^, but not in the presence of Ca^2+^ [162]. Voltage-dependent K^+^ channels are present in contractile vacuoles [167]. 

*D. discoideum* possesses two types of proton-driven metal ion transporters from the Nramp superfamily: Nramp1/Nramp2, which are conserved from bacteria to humans [168]. In contrast to Nramp1, which is localized to phagosomes and macropinosomes, Nramp2 is only found in the membrane of contractile vacuoles. Both proteins work together with the proton ATPase. Disrupting both genes leads to developmental defects such as delayed cell aggregation, suggesting that Nramp1/2-regulated Fe^2+^ homeostasis is involved in cell development. Nramp1 also provides resistance against infection by invading bacteria, much like its mammalian counterpart [168]. 

The resting membrane potential in the *D. discoideum* plasma membrane is significantly influenced by the proton ATPase [120], and is estimated to be approximately −46 mV [169]. While membrane potential maintenance significantly affects galvanotaxis, it does not influence chemotaxis [170,171]. However, the contribution of membrane charge states to cell polarity formation during chemotaxis is gaining increased recognition [172], and further clarification is needed on how cells selectively process electrical signal information.

## 3. Conclusions and Perspectives

The fluctuating dynamics of intracellular ions can be determined using electrodes and radioisotopes, and were measured before the development of fluorescent probes [151,154,155,156,163]. However, most measurements using these techniques are made in multiple cells, such as cell suspensions, and thus cannot be extrapolated to the single-cell level. Because biological phenomena exhibit probability distributions, inferring single-cell or single-protein level functions from measurements based on bulk averages is complicated. The development of fluorescent probes based on fluorescent proteins [30,173] has made it possible to selectively and microscopically capture ion dynamics at the single-cell level [25,42,130,174]. In the future, high spatiotemporal resolution measurements at the single-cell local level combined with super-resolution microscopy and other techniques are expected to visualize the spatial dynamics of ion signals and elucidate the molecular mechanisms of ion signaling.

While this review focuses on understanding ion signaling in *D. discoideum*, much of what has been revealed in *D. discoideum* cells contributes to our understanding of general cell biology. *D. discoideum*, which is easily cultured and manipulated, is an excellent model organism for studying signal transduction and cell development. Additionally, verifying whether mechanisms in mammalian cells or other microbial cells are also consistent in cellular slime molds is crucial to establish an evolutionary view of cells in general. As signal switching occurs between the unicellular and multicellular phases within the same *D. discoideum* strain [22,25], it provides powerful evidence of what is essentially different between unicellular and multicellular systems. The ability to obtain accurate measurements using *D. discoideum* cells, which have long been studied as a model organism for signal transduction, and to compare them with older data will greatly contribute to organizing basic biological knowledge.

## Figures and Tables

**Figure 1 biomolecules-14-00830-f001:**
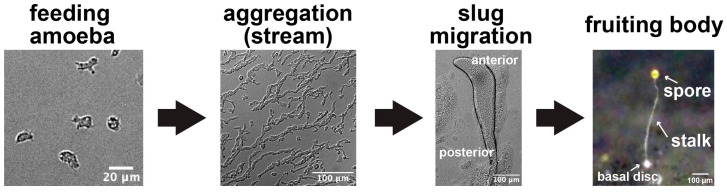
Life cycle of *Dictyostelium discoideum*. During the unicellular phase, cells proliferate as amoebae by feeding on bacteria. When the cells become starved, they form streams and aggregate due to the chemotaxis toward the cAMP they produce. The cells form a “slug”, which is a multicellular body consisting of approximately 100,000 cells, and begin to migrate. Eventually, they form a fruiting body comprising spores and a stalk. The spores germinate and revert to amoeboid cells upon reaching a suitable environment.

**Figure 2 biomolecules-14-00830-f002:**
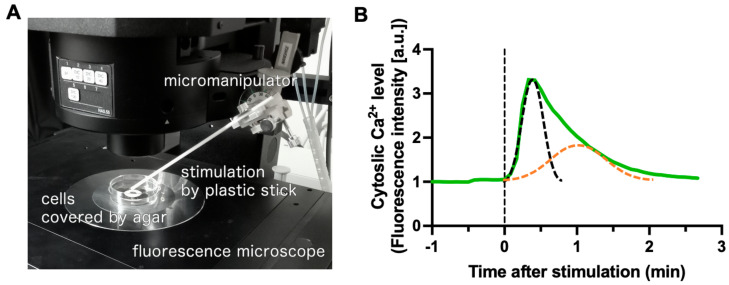
Calcium signaling in response to mechanical stimuli in a *Dictyostelium discoideum* slug. (**A**) A slug of *D. discoideum* cells expressing the calcium probe, GCaMP6s, was placed between a coverslip and an agar sheet and subjected to mechanical pressure by pushing the agar sheet with a plastic stick. The fluorescence signal of GCaMP6s was observed under a fluorescence microscope. (**B**) A representative time course of normalized fluorescence intensity of GCaMP6s in a slug after mechanical stimulation (solid green line), demonstrating an early peak dependent on the extracellular Ca^2+^ influx (dashed black line) and a later peak dependent on the intracellular vesicles’ Ca^2+^ flux (dashed orange line) [25].

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
