# Peer review of "Ion Signaling in Cell Motility and Development in Dictyostelium discoideum"

_biomolecules, 2024, doi:10.3390/biom14070830_

Round 1

Reviewer 1 Report

Comments and Suggestions for Authors

This review brings together information on a number of ions implicated in either cell movement or development in the social amoeba Dictyostelium discoideum. It is valuable to bring all of this information together in one place and provides an concise overview of what is known about the roles of Ca2+, H+ and Na+/K+, including the methods used to measure these.

1.      The last sentence of the abstract is not a full sentence and needs rewriting

P1 line 41 ‘selective ion transfer by ion channels is necessary to constitute the concentration gradient’ does not make sense. The gradient is set up by energy-dependent pumps and ions flow though channels down their gradient. Both of these are specific for particular ions.

1.      P2 line 71-72. D. discoideum has been studied not just because of the ease of culture. Importantly in this context, conservation of proteins, pathways and processes  with higher eukaryotes, along with ease of genetic manipulation and a  simplified multicellular lifecycle are important to mention.

2.    Lines 102-4 Should include DIF1 and DIF2 in this list, Traynor and Kay Biol Open 2017 Feb 15;6(2):200-209

3.        P3 line 106. This sentence ‘This ATP

and ADP-mediated calcium signaling has also been implicated in P2 receptors localized

in the contractile vacuole ‘does not

make sense and needs rewriting. There are a family of P2X receptors in D. discoideum, which have been implicated in  Ca2+ signalling and osmoregulation. Should also reference Parkinson et al Nat Cell Biol. 2014;16:87-98.

4.      P3. The section from line 111 to 117 relates to the role of extracellular Ca2+ and is mixed in with discussion of intracellular Ca2+ cytosolic rises on either side of this section.  This section on extracellular Ca2+ would be better moved to the end of the paragraph or in a paragraph of its own.

5.      P3 Line 117 it is not just the dynamics of the Ca2+ oscillations but any detectable Ca2+ rise at all that depends on ipla. This needs clarifying.

6.      P3. Somewhere it should be explained that iplA is the only Dictyostelium orthologue of the mammalian IP3 receptor, a ligand-gated Ca2+ channel for release of Ca2+ from ER stores.

7.      P3 122-3 this should read ‘Strains deficient in RegA, a cAMP phosphodiesterase, ,…. the role of cAMP in controlling cytosolic Ca2+ levels’

8.      P3 Line 126-7 Loss of TPC2 has been shown to effect D. discoideum development but this has not been proven to work through altered Ca2+ signalling. This sentence needs re-writing to be correct. In fact the change in Ca2+ response to cAMP in the tpc2null cells is quite minor (Chang et al).

9.      Needs some more explanation about what TPC channels are for those who are not familiar with these – where they are found, what gates them and which Ions can pass through them. There is evidence that these can be Na+ channels in some cases.

10. P3 134-5 should also reference Traynor and Kay here. Biol Open. 2017;6(2):200-209

11. P4 144-5. DdCAD adhesion depends on Ca2+ outside the cell for adhesion – whereas everything else discussed is intracellular. This should be explained or the sentence on cadherin removed.

12. Line151 and 154. Include a brief explanation of Trp channels and Piezo when these proteins are first mentioned.

13. P5 line 182 Should read ‘Ca2+ influx and efflux were measured using radioisotopes’ – can only measure fluxes.

14. P5 line 185. Should point out that the sensitivity of aequorin is not ideal for Dictyostelium cells which have a lower range of cytosolic Ca2+ than mammalian cells, until late in development when basal cytosolic Ca2+ levels rise.

15. P5 line 206. I should be explained that proton ATPases use ATP to provide energy to pump protons.

16. P5 line 222. A much more recent paper on cytosolic pH and cell differentiation in Dictyostelium is Gruenheit et al Dev Cell. 2018 Nov 19;47(4):494-508 and should be included. All the refrences cited here are old, which is fine, but there are more recent ones.

17. Line 222-3 This is Gruenheit et al reference should also be included when discussing the role of cytosolic pH in gene expression as it includes RNAseq data from cells grown in different pH media

18. Weak bases like ammonia will alter the pH of acidic vesicles rather than the cytosol and this should be explained. Davies L, Satre M, Martin JB, Gross JD.Cell. 1993 Oct 22;75(2):321-7.

19. Fe ions also have been implicated in Dictyostelium development as deletion of two iron transporters (nramp1/2) leads to developmental defects. These null cells also show increased sensitivity to some bacterial infections. For the sake of completion, this should be included. See papers by Bozzaro et al ,especially Peracino B, Buracco S, Bozzaro S.J Cell Sci. 2013 ;126:301-11 for developmental effects.

Comments on the Quality of English Language

Minor editing of English language required

Reviewer 2 Report

Comments and Suggestions for Authors

After two expert opinions had already been received, I was asked for a further short statement. In my opinion, this is a valuable review and a helpful resource for all researchers who are particularly interested in calcium signaling in organisms beyond animals. I recommend publication without further changes.

Reviewer 3 Report

Comments and Suggestions for Authors

In the manuscript, Dr. Morimoto discusses the role of ion signaling in the mobility and development of Dictyostelium discoideum cells. This organism, Dictyostelium, is a promising model system that can transition from a unicellular to multicellular state during its life cycle, making it convenient for studying the role of molecular components in development. Understanding the molecular mechanisms that drive these processes in Dictyostelium can provide insight into the developmental pathways of more complex organisms, such as humans, and contribute to our understanding of oncogenesis. The review summarizes the latest experimental findings on this topic, including those from Morimoto and his team. The review focuses on various aspects of Dictyostelium biology, emphasizing the role of ions, especially calcium, in cellular processes. It provides a comprehensive overview of the current state of research on this organism and its potential applications in understanding developmental biology and oncogenesis.It also provides a brief overview of the latest methods that allow controlling the concentration of various ions in cellular processes. This review will be useful and interesting for a wide range of readers, especially researchers studying the process of cell differentiation during the development of multicellular organisms. The manuscript is well-written and the experiments are well-conducted. Several points require attention before publication. These are listed below in the order in which they occurred while reading the manuscript:

Abstract, lines 23-24. “Here, I review the role of ion signaling in D. discoideum cell- to-cell”.

This phrase is very brief, could you please provide more detail to better reflect the contents of the review?

Figure 1, line 75. The fourth image, which shows the fruiting body, lacks a scale bar.

The two phrases Line 107: “While Ca2+ signals are not essential for chemotaxis [51]…   and Lines 115-117: “Calcium chemotaxis to Ca2+ concentration gradients is also observed with contributions from the IP3 receptor homolog, IplA and myosin heavy chain kinase [56,66]” seem to contradict each other to me. Can you please explain?

Lines 211-212: “An increase in intracellular pH enhances cell migration, while a decrease reduces migration [117]” - Could you please clarify the range of pH changes that you are referring to?

Lines 213-215. “Intracellular pH is regulated by the Na+/H+ exchanger Nhe1, suggesting that pH is elevated at the polarized leading edge of migrating cells [130]”. Could you please explain this finding in more detail?

Line 232: “pH of the vacuoles in prestalk cells is notably low [148,149]” - Please clarify what pH values are meant.

Lines 242-243. Please use the standardized name for this enzyme, Nhe1 or NHE. Could you please explain why the relationship between this enzyme and pH is significant in cancer cell research?

Reviewer 4 Report

Comments and Suggestions for Authors

This is an important attempt at bringing together information on signaling in Dictyostelium discoideum.  While it is well written for the most part (I provide a few suggestions below) I find it hard to picture how all the signaling works.  I have some knowledge of this organism, but I am trying to read this through eyes of the non-expert.

For example, section 2.1.1

Lines 96-97 How does Ca increase in response to cAMP relay?

Lines 99-106  It looks everything increases intracellular Ca.  How is all of this happening?  How are signals kept separate and interpreted by the cell?

Another example, Section 2.3

Lines 252-255   Within the space of two sentences, we are told that K concentration associated with cAMP signaling is suggested and then K effects on differentiation has been implied.  Short vague sentences like these are not helpful to the reader. More specificity would improve the discussion.

Line 20  delete “offers”

Line 31 changed “perceived” to “transduced into”

Line 39  change “serving as” to “such as”

Comments on the Quality of English Language
